# Beneficial Effects of Inflammatory Cytokine-Targeting Aptamers in an Animal Model of Chronic Prostatitis

**DOI:** 10.3390/ijms21113953

**Published:** 2020-05-31

**Authors:** Dong-Ru Ho, Pey-Jium Chang, Wei-Yu Lin, Yun-Ching Huang, Jian-Hui Lin, Kuo-Tsai Huang, Wai-Nga Chan, Chih-Shou Chen

**Affiliations:** 1Division of Urology, Department of Surgery, Chang Gung Memorial Hospital, Chiayi 613016, Taiwan; redoxdrh@gmail.com (D.-R.H.); checotrade@gmail.com (W.-Y.L.); abradva@gmail.com (Y.-C.H.); d9400404@stmail.cgu.edu.tw (J.-H.L.); ronsolglobalinc@gmail.com (K.-T.H.); yogochencjh@gmail.com (W.-N.C.); 2Graduate Institute of Clinical Medical Sciences, College of Medicine, Chang Gung University, Taoyuan City 333323, Taiwan; r91a21015@ntu.edu.tw; 3Department of Nursing, Chang Gung University of Science and Technology, Chiayi 613016, Taiwan; 4Department of Medicine, College of Medicine, Chang Gung University, Taoyuan City 333323, Taiwan

**Keywords:** inflammasome, aptamer, chronic prostatitis, chronic pelvic pain syndrome, interleukin-1 beta, tumor necrosis factor-alpha, apoptosis

## Abstract

Non-bacterial prostatitis is an inflammatory disease that is difficult to treat. Oligonucleotide aptamers are well known for their stability and flexibility in conjugating various inflammatory molecules. In this study, we investigated the effects of inflammatory cytokine-targeting aptamers (ICTA), putative neutralizers of TNF-alpha and IL-1 beta activation, on local carrageenan-induced prostate inflammation, allodynia, and hyperalgesia in rats. In vitro evaluation confirmed the binding capability of ICTA. Intraprostatic injection of carrageenan or control vehicle was performed in six-week-old rats, and ICTA (150 µg) or vehicle was administered in the prostate along with carrageenan injection. The von Frey filament test was performed to determine mechanical allodynia, and prostate inflammation was examined seven days after drug administration. Local carrageenan administration resulted in a reduction of the tactile threshold. The levels of mononuclear cell infiltration, pro-inflammatory cytokine interleukin-1 beta (b), caspase-1 (casp-1), and Nucleotide-binding oligomerization domain, Leucine rich Repeat and Pyrin domain containing proteins 1 and 3 (NALP1 and NALP3) in the prostate of rats were increased seven days after carrageenan injection. Treatment with ICTA significantly attenuated the carrageenan-induced hyperalgesia and reduced the elevated levels of proteins including TNF-a and IL-1b in the rats. Apoptosis markers, B-cell lymphoma 2-associated X protein (Bax) and caspase-3, were elevated in ICTA-treated Chronic pelvic pain syndrome (CPPS) rats. These results suggest that ICTA provides protection against local carrageenan-induced enhanced pain sensitivity, and that the neutralization of proinflammatory cytokines may result in inflammatory cell apoptosis.

## 1. Introduction

Chronic prostatitis (CP), also known as chronic pelvic pain syndrome (CPPS, non-bacterial prostatitis, or type III prostatitis by National Institutes of Health consensus) is a form of inflammation that can potentially induce long-lasting alteration in pain sensitivity [1]. The inflammatory hyperalgesia may be related to adrenocortical hormone [2], mast cells [3], or myofascial pain [4]. Clinical symptoms were found to be related to the imbalance between proinflammatory and anti-inflammatory cytokines [5], including IL-8, IL-10 [6], IL-1b, and TNF-a [7]. Therefore, the treatment or prevention of further neurological damage induced by inflammation may be able to effectively reduce the incidence of pain.

In our previous study, we indicated that the inflammasome may play an important role in non-bacterial prostate inflammation [8], as also suggested in other studies [9,10,11]. In addition, the inflammasome has gradually become a target for treating pain [12] and inflammation [13,14]. However, there is still no definitive treatment targeting the inflammasome in CPPS. Additional studies indicated that the existence of IL-1b and TNF-a in prostatic secretions is associated with hyperalgesia [15], which is possibly mediated by the infiltration of CD4^+^ cells [16]. However, the relationship between hyperalgesia improvement and inflammasome blockade following local inflammation in the prostate is unclear. 

Evidence shows that IL-1b neutralization with monoclonal antibodies has potential for treating inflammatory diseases [17,18,19]. It has also been reported that TNF-a-targeting aptamer has anti-inflammatory effects and has been proven to reduce TNF-a-mediated acute lung injury and acute liver failure [20]. Neutralizing inflammatory cytokines can be of benefit in various diseases. Moreover, inflammatory cytokines may cross-react with one another. A previous study revealed that IL-1b augments TNF-alpha-mediated inflammatory responses in lung epithelial cells [21]. IL-1b pretreatment enhanced TNF-alpha-induced macrophage inflammatory protein in a post-transcriptional manner. Thus, to prevent inflammatory pain induced by carrageenan, it may be best to block the effects of major cytokines. However, the mechanism behind the effects of aptamers on carrageenan-induced pain sensitivity requires further study.

Our previous data indicated that intra-prostatic (i.p.) injection of carrageenan resulted in hyperalgesia 48 h after drug administration and continued for seven days in rats. In addition, this hyperalgesia induced by i.p. carrageenan could be reduced by blocking inflammasome activation with chlorogenic acid. However, it is not clear that blocking end products of the inflammasome pathway helps to stop inflammation.

Therefore, in the present study, we developed a novel IL-1b-targeting aptamer (AptIL-1b) combined with TNF-a aptamer (AptTNF-a) to form inflammatory cytokine-targeting aptamers (ICTA) to promote maximal cytokine neutralization. The objectives of this study are to determine (1) the therapeutic effect of ICTA on prostate hyperalgesia and (2) the possible inflammatory factors through which ICTA provides protective effects against subsequent carrageenan-induced prostate inflammation.

## 2. Results

### 2.1. AptIL-1b Binds to IL-1b With High Affinity and Targets IL-1b In Vitro

An IL-1b-targeting aptamer (AptIL-1b, Figure 1A) was selected using the nitrocellulose filter SELEX and optimized based on the predicted secondary structure using Mfold. AptTNF-a was adapted from the study by Lai et al. [20]. AptIL-1b and AptTNF-a can bind to rat IL-1b and TNF-a respectively (Figure 1B,C). As the data further showed that there is a dose response of AptIL-1b binding to human IL-1b, as demonstrated by enzyme-linked immunosorbent assay (ELISA) (Figure 1D), we subsequently investigated the in vitro effects of AptIL-1b using the Jurkat cell line (Figure 1E) and AptTNF-a using a urothelial cell line (Figure 1F). Our data showed that IL-1b clearly stimulated cell proliferation after 48 h and AptIL-1b counteracted its effects. In addition, TNF-a caused a cytotoxic effect after 48 h, which was counteracted by AptTNF-a. In contrast, no significant changes were found upon treatment with AptIL-1b or AptTNF-a alone.

### 2.2. ICTA Attenuated Local Carrageenan-Induced Mechanical Allodynia and Tactile Hyperalgesia 

All male rats received intraprostatic injection with control vehicle, carrageenan (CPPS group), carrageenan with ICTA (CPPS + ICTA group), or ICTA alone. Mechanical allodynia was assessed using the von Frey filament test at both the scrotal wall and the tail base. Consistent with our previous study, local carrageenan intraprostatic injection to SD rats resulted in a lower pain threshold at the scrotal wall of SD rats compared with that of the control group (*p* < 0.05) (Figure 2A). ICTA treatment significantly reduced the carrageenan-induced mechanical allodynia in SD rats (*p* < 0.05) (Figure 2A). The pain threshold at the tail base was not reduced significantly (Figure 2B).

### 2.3. ICTA Modified Local Carrageenan-Induced Glandular Hyperplasia and Inflammatory Responses in the Prostate

Local carrageenan treatment stimulated hyperplasia of glandular epithelium in the prostate as indicated by Haemotoxylin& Eosin (H&E) staining of 7-µm-thick slices, which was determined in the ventral prostate sections of the male SD rats (Figure 3). In the control rat prostate (Figure 3A), flat epithelium was detected, and most of those cells were in a monolayer state with a cuboidal shape, supranuclear clear areas, and brush border (arrows). When carrageenan induced CPPS, we found significantly increased numbers of interstitial mononuclear cells, which is an indication of macrophage (arrows) and lymphocyte (arrowhead) activation. This was found in the carrageenan-treated prostate of the rats 168 h after carrageenan injection (Figure 3B). ICTA treatment did not reduce the number of infiltrated mononuclear cells (arrows) in the interstitium of the ventral prostate. In fact, cell numbers increased by 27% in the CPPS + ICTA group (Figure 3E). Caspase-1 expression significantly increased in the CPPS + ICTA group (Figure 3T), but the expression of TNF-a in this group was significantly lower than that in the CPPS group (*p* < 0.05) (Figure 3O). IL-1b expression was lowered in CPPS + ICTA group compared to CPPS group, but not significantly (Figure 3J).

Local exposure to carrageenan resulted in inflammasome activation in rat prostate, as evidenced by the increases in NALP1^+^ cells (Figure 4B) and NALP3^+^ cells (Figure 4G). Seven days after carrageenan injection, NALP1 and NALP3 protein expression in the prostate of carrageenan-exposed rats (CPPS) dramatically increased compared with that in the saline-injected control rats (*p* < 0.05) (Figure 4A,F). ICTA attenuated the carrageenan-induced increase in the concentration levels of NALP1 in the prostate of rats (*p* < 0.05) (Figure 4E). However, NALP3 expression in the CPPS + ICTA group prostate was significantly increased compared with that in the control and CPPS groups (*p* < 0.05) (Figure 4J). ICTA alone had little effect on either NALP1 or NALP3 expression in the prostate (Figure 4D,I).

### 2.4. ICTA Promoted Local Carrageenan-Induced Apoptosis in the Prostate

Local exposure to carrageenan not only induced inflammasome activation, but also promoted the expression of caspase-3, which is an indicator of apoptosis (Figure 3B). ICTA neutralized the end products of the inflammasome (IL-1b) and related cytokines (TNF-a), but did not stop the inflammasome pathway from becoming activated. This may have resulted in cells with an activated inflammasome pathway becoming stressed and undergoing apoptosis. ICTA did promote BAX- and Caspase-3-expressing infiltrative mononuclear cells in the interstitial space of prostate following local carrageenan injection (*p* < 0.05) upon treating CPPS (Figure 5C,H). The results are expressed as the mean ± SD of six rats in each group and were analyzed by Student’s *t*-test. * *p* < 0.05 represents significance for the CPPS group or CPPS + ICTA group compared with the control group. ** *p* < 0.05 represents significance for the CPPS + ICTA group compared with the CPPS group (*n* = 6).

The active key inflammatory mediator IL-1b, which is excreted extracellularly in the original form pro-IL-1b, was regulated in rat prostate through inflammasome pathway NALP1&3 and caspase-1 following local carrageenan exposure (Figure 3G). When neutralizing ICTA was injected simultaneously, the concentration level of intracellular caspase-1 (Figure 3R,T) continued to increase along with NALP3 (Figure 4H,J) compared with that in the control group (*p* < 0.05). ICTA attenuated the effects of IL-1b and TNF-a, but did not stop the increase in carrageenan-induced inflammasome activation. Significant increases in caspase-3 (Figure 5C) and BAX (Figure 5H), both apoptosis-related proteins, were found in cells infiltrated into the interstitial space of the prostate (*p* < 0.05) (Figure 5E,J).

## 3. Discussion

Chronic prostatitis (CP)-related chronic pelvic pain syndrome (CPPS) is poorly understood and undertreated. Prostatitis was categorized by the National Institutes of Health, in which type I and II prostatitis result from identifiable prostatic infections, and type IV is asymptomatic. The majority of symptomatic cases are in type III, also named chronic prostatitis (CP/CPPS) [22]. Patients with CP/CPPS typically report genital or pelvic pain (penis, perineum, scrotum) lasting more than 3 months [23]. Localized prostate uropathogens were reported in only 6–8% of CP/CPPS patients [24]. It is believed that immune and endocrine alteration is what causes changes in neural function [25].

### 3.1. Severity of Clinical Symptoms is Related to the Inflammasome Pathway

Our data showed that local carrageenan injections induced mechanical allodynia and hyperalgesia in SD rats [8]. These pain responses in the scrotal wall and tail base may be correlated with the severity of local prostate inflammation [26]. The severity of CPPS is correlated with IL-1b and TNF-a levels in prostate excretions. Vascular endothelial growth factor A, interleukin 6 (IL-6), IL-1b, TNF-a, and cytochrome c oxidase subunit II are correlated with the severity of CP/CPPS [27]. Cytokines, namely IL-1b and TNF-a, are frequently present and elevated in the EPS from men with CPPS, and provide a novel means of identifying, characterizing, and potential managing men with this condition. When local cytokine neutralization was achieved by aptamers targeting IL-1b and TNF-a, this gave rise to an increased nociceptive threshold related to changes of cytokine production, inflammasome pathway (type 1& 3 Nucleotide-binding oligomerization domain, Leucine rich Repeat and Pyrin domain containing (NALP1 &3) and Caspase-1) activation, and apoptosis initiation, for example, BAX and Caspase-3, as summarized in Figure 6. Taking these findings together, blocking local inflammation with a cytokine-targeting agent, ICTA, might stop the inflammatory cascade of infiltrated mononuclear cells in the ventral prostate and, as a result, promote the apoptosis of these cells and reduce pain in SD rats.

### 3.2. Inflammasome is Important in CP/CPPS

Inflammasome activation in the prostate involves a complex interaction between danger-associated molecules (DAMPs) and glandular endothelium, macrophages, or lymphocytes [28,29], among others. It has been reported that inflammasomes can be seen in prostate epithelial cells and infiltrating macrophages [30]. Pathogen infection [11], hormone imbalance [31], hypoxia [30], and chemicals [32] are all factors that can result in inflammasomes. NALP1 [8,32], NALP3 [10,11,31,33], and AIM2 [9,30] inflammasomes were found important in prostate diseases. However, caspase-1 may even be activated by TNF-a in an NALP1- and NALP3-dispensable manner [34]. The present results showed that hyperalgesia and allodynia induced by local inflammation are closely associated with increases in NALP1 and NALP3 activation, as well as pro-inflammatory cytokines IL-1b and TNF-a in the ventral prostate of rats.

### 3.3. Cytokines Responsible for Inflammation

Cytokines have been shown to be the targets of many inflammatory diseases. One common treatment modality is introducing neutralizing monoclonal antibody, which may block the effects of cytokines and reduce inflammatory cascades. Blocking IL-1 activity has been used to treat autoinflammatory syndromes and would result in a rapid and sustained reduction in disease severity. Currently approved IL-1-targeted agents include the IL-1 receptor antagonist anakinra, the soluble decoy receptor rilonacept, and the neutralizing monoclonal anti-IL-1β antibody canakinumab [17]. Neutralization of pro-inflammatory cytokines, especially IL-1b with the IL-1 receptor antagonist (IL-1Ra) and/or IL-1b antibodies, has been found to result in abolition of the inflammatory process of pancreatic islets and consequently a decrease in insulin resistance [35].

TNF-a is also a treatment target of inflammatory diseases. Anti–TNF-a agents have been widely used in Crohn’s disease and other autoimmune-mediated inflammatory diseases, such as rheumatoid and psoriatic arthritis, psoriasis, and ankylosing spondylitis [36]. Monoclonal antibodies targeting TNF-a, including infliximab (Remicade), adalimumab (Humira), and certolizumab pegol, have all been shown to be effective in inflammatory bowel diseases. Other inflammatory cytokine-targeting modalities include anti-IL-6 receptor antibodies and TH17-targeting antibodies [37]. In some studies, the blocking of IL-1b or TNF-a alone was shown to be sufficient for disease control [36,38]. In other cases, inflammation needed to be reduced by cotreatment with TNFα and IL-1 antagonists, namely, etanercept and Kineret, respectively. This indicates that IL-1 frequently acts in concert with TNFα in certain inflammatory processes [39]. Based on these studies [7,21,39], we evaluated the roles of IL-1b and TNF-a, the major cytokines of CP/CPPS, in regulating the inflammasome pathway associated with carrageenan-induced inflammatory reactions at both prostate interstitium and glandular epithelium in our animal model. The results indicated that blocking of the main cytokines simultaneously is beneficial for local prostate inflammation, and therefore an ICTA protocol for neutralizing both cytokines was designed.

### 3.4. Besides Cytokines, Inflammasome is Also a Good Target for Treating Inflammation

Recent studies targeted the NALP1 inflammasome [40], IL-6 [41], and chemotactic protein [42] in efforts to significantly reduce inflammation in brain injury, myocardiac infarction, and pulmonary emphysema, respectively. In addition, anti-Apoptosis-associated Speck-Like Protein Containing a Caspase Recruitment Domain (ASC)-neutralizing antibodies administered immediately after traumatic brain injury in rats were shown to potentially reduce caspase-1 activation and the processing of IL-1b. This was proven to significantly decrease brain contusion volume [40]. These studies demonstrated that, if the inflammasome pathway is managed well at a certain period of inflammation, the vicious cycle may be reversed.

### 3.5. Aptamers Were Used to Detect and Treat Inflammation-Related Diseases

Aptamers are oligonucleotide or peptide molecules that bind to a specific target molecule. They have been applied to attenuate inflammatory organ damage, neutralizing the endotoxicity of LPS, and improving survival in multiorgan failure models [43,44]. However, they can also work for local inflammation. For example, aptamers blocking IL-17 binding to IL-17RA can relieve synovial destruction in animal arthritis models [45]. They can also work on upstream initiators such as NF-kB. One study successfully introduced aptamers into cells to inhibit inflammatory responses induced by TNF-a via downregulation of the nuclear NF-kB subunit [46]. Although there are numerous inflammatory cytokines, only 11 aptamers have been created against either cytokines or their receptors. These cytokines include IFN-gamma, IL-2, IL-6, IL-10, IL-11, IL-17, IL-32, IP-10, CCL2, TGF-b, and TNF-a. Most of the isolated aptamers target pro-inflammatory or dual-function cytokines, and they could be used for the diagnosis, treatment, or prevention of inflammatory diseases [47]. The aptamers (ICTA) in our study combine the newly discovered IL-1b aptamers and the TNF-a aptamer from the study by Lai et al. [20]. *In vitro* studies first demonstrated the effect of ICTA on cytokine conjugation. In addition, the *in vivo* treatment with local injection demonstrated a benefit over systemic use. Although there are already well-known monoclonal antibodies for neutralizing cytokines in inflammatory diseases, special transport and storage facilities may be needed to apply these drugs in clinical practice. Therefore, we developed ICTA based on single-stranded DNA oligonucleotides to meet this need.

### 3.6. Apoptosis in Inflammation

Clinical and animal research studies have indicated that the apoptosis of inflammatory cells is related to the cessation of cytokine-related diseases. Anti-TNF-a antibodies, such as adalimumab and infliximab, have been proven to affect the cytokine production of monocytes and induce their apoptosis [48]. This is related to a decreased level of soluble tumor necrosis factor in cell culture. These antibodies were also found to induce T-cell apoptosis in inflammatory bowel diseases [49], which may be related indirectly to the mTNF/TNFR2 pathway. IL-1 blockade with canakinumab was also found to increase the apoptosis of neutrophils and decrease pro-inflammatory signaling. This may be related to MAPK14 downregulation, which is in the downstream pathway of the drug target IL-1β [50]. In the research on inflammatory cytokine-related cachexia, TNF-a, IL-1, IL-6, and IFN-gamma were all suggested to be related to cell death along with chronic diseases [51]. Besides cytokines, caspase-1-induced apoptosis has been noted in macrophages with activation of the Bid-caspase-9-caspase-3 axis, which can be followed by secondary necrosis/pyroptosis [52]. In our study, inflammatory cells in ICTA-treated CPPS rats also demonstrated overexpression of caspase-1.

William et al. commented that the mechanism through which an IL-1-expressing cell dies dictates the nature of the inflammation. This allows modification of inflammation through the selective targeting of cell death mechanisms during disease [53]. Generally speaking, apoptotic cells will induce anti-inflammatory responses, and thus apoptotic cell removal is usually immunologically silent [54]. Many anti-inflammatory mechanisms initiated by apoptotic cells have been found, including the release of anti-inflammatory molecules from them, immediate anti-inflammatory signaling pathways triggered by apoptotic cell surface molecules, and the production of anti-inflammatory soluble mediators by phagocytes [55]. These findings all indicate that apoptosis is one of the measures to modulate inflammation. Further studies should investigate the cause of apoptosis activation, for which endoplasmic reticulum stress may be key. Although inflammasome activation has been found to be the result of endoplasmic reticulum stress, it may in turn bring more of a burden if feedback regulation is lost.

### 3.7. Concomitant Neutralization of Cytokines

Sometimes, only concomitant neutralization of cytokines helps to stop inflammation. It has been proven in a study of pneumococcal infection following influenza that concomitant neutralization of IFN-r and IL-6 significantly reduced the severity of pneumonia and bacteremia [56]. Another example in NK cells with co-stimulation of IL-2 receptor and IL-12 receptor induced significant IFN-r production, but this was followed by NK cell apoptosis and a decline in IFN-gamma production [57]. Therefore, we decided to use ICTA targeting both major cytokines in CP/CPPS to achieve the maximal treatment effect. Our results demonstrate that concomitant neutralization of IL-1b and TNF-a with ICTA immediately after the inflammatory stimulation can reduce inflammation and pain behavioral responses through inducing apoptosis in inflammatory cells, and therefore prevent the vicious circle of local prostatitis progression. 

### 3.8. Potential Problems and Research Topics With ICTA Administration

There may be more than two inflammatory cytokines activated in response to *in vivo* carrageenan stimuli. Neutralizing only IL-1b and TNF-a may not be enough to block the entire inflammasome pathway, but it could demonstrate physiological and pathological changes in this study. However, there are still some problems associated with applying ICTA to inflammatory diseases. First, NF-kB was initially evaluated in our experiments, but the variation among rats was so large that it was difficult to pinpoint its role under the current study design. NF-kB has been noted for its complex roles in inflammation that affect both pro- and anti-inflammatory pathways [58], so its role in ICTA administration will need to be clarified. Second, other cytokines not directly associated with inflammasome, such as IL-18, were expressed with a wide range of variation in this study. Because relationships between pro- and anti-inflammatory cytokine genes can change depending on the biological conditions [59], cytokine expression profiles will be an interesting topic for future study when applying ICTA to treat inflammatory diseases.

## 4. Materials and Methods 

### 4.1. Oligonucleotides

Unless otherwise stated, all chemicals used in this study were purchased from Sigma (St. Louis, MO, USA). All oligonucleotides were synthesized by Mingshin Biotechnologies (Taipei, Taiwan R.O.C.). The sequence of IL-1b-targeting aptamer (AptIL-1b) is 5’-GAAGT ATAGC GCAAT AAACT CACCT CCCAC-3’ and that of TNF-a-targeting aptamer (AptTNF-a) is 5’-GCGCC ACTAC AGGGG AGCTG CCATTC GAATA GGTGG GCCGC-3’. ICTA used in the animal study is composed of 150 µg of AptIL-1b and 150 µg of AptTNF-a in 150 µL of PBS for each injection.

### 4.2. Construction of IL-1b-Targeting Aptamer

Rat IL-1b-targeting aptamers were screened using the nitrocellulose filter Selective Evolution of Ligands by EXponential enrichment (SELEX). The synthetic single-stranded DNA library contains 70-nucleotide-long single-stranded DNAs with 30 random sequences flanked by primer sequences, 5’-TAGGG AAGAG AAGGA CATAT GAT [N]_30_ TTGAC TAGTA CATGA CCACT TGA, N = A, T, G, C. In the first SELEX round, the 10^15^-molecule ssDNA library was incubated with recombinant rat IL-1b proteins (R&D Systems, Inc., Minneapolis, MN, USA). The ssDNAs that bound to IL-1b proteins were picked up by the nitrocellulose filter and the unbound ssDNAs were removed through repeated washing. The IL-1b-bound ssDNAs were then eluted by heating, negatively selected by incubating with albumin, and then filtered through the nitrocellulose membrane. The eluted solution was collected and treated with PCR. The SELEX was repeated for 10 rounds. Both IL-1b-bound ssDNAs and the albumin-bound ssDNAs were sent for next-generation sequencing (Illumina MiSeq System). The output reads in the IL-1b-bound ssDNA group were clustered by FASTApatmer software (Bond Life Sciences Center, University of Missouri, USA) [28] and subtracted with the clusters that appeared in the albumin-bound group. The highest-read sequences’ structures were analyzed using Mfold (The RNA Institute, College of Arts and Sciences, State University of New York at Albany). The secondary structures were further predicted.

### 4.3. Antibodies

Monoclonal mouse antibodies against TNF-a (sc-80383), b-actin (sc-47778), NF-kB (sc-8008), and pNF-kB (sc-135768) and polyclonal rabbit antibodies against IL-1b, (sc-7884), NALP1 (sc-66846), NALP3 (sc-66992), caspase-1 (sc-515), BAX (sc-493), Caspase-3 (sc-7148) cleaved caspase-3, and anti-glyceraldehyde-3-phosphate dehydrogenase (GAPDH, sc-20357) were purchased from Santa Cruz Biotechnology (Santa Cruz, CA, USA). ELISA kits for immunoassays of rat IL-1b and TNF-a were purchased from R&D Systems (Minneapolis, MN, USA).

### 4.4. Animals

Sprague Dawley rats arrived at the laboratory at 4 weeks of age. They were maintained in an animal room on a 12-h light/dark cycle and at constant temperature (22 to 25 °C). All procedures for animal care were approved by the Institutional Animal Care and Use Committee at the Chang Gung Memorial Hospital Laboratory Animal Center (#2017092801; approval on 17 December 2017). Every effort was made to minimize the number of animals used and their suffering.

### 4.5. Animal Treatment

The rats were divided into four groups: Control (saline + vehicle, *n* = 6), CPPS (carrageenan + vehicle, *n* = 8), CPPS + ICTA (carrageenan + ICTA, *n* = 6), and (saline + ICTA, *n* = 6). Intraprostatic injection of carrageenan (0.75 mg/0.2 mL) or sterile saline (total volume of 0.2 mL) in 6 week-old Sprague–Dawley male rats was performed as previously described [8]. Then, 150 µg/200 gm of ICTA in phosphate-buffered saline (PBS) or PBS alone with a total volume of 0.1 mL was injected 5 min after the carrageenan injection [17]. Seven days after the injection, rats from each group were sacrificed by carbon dioxide euthanasia followed by 4% paraformaldehyde for preparing prostate sections. Consecutive frozen ventral prostate sections at a thickness of 7 µm were prepared in a cryostat for immunohistochemical examination.

### 4.6. Von Frey Filament Test

The von Frey filament test is used to evaluate the mechanical nociception (skin mechanical sensitivity) of rats. This test was completed as described in our previous study with modifications on preoperative day 1 and 5 days after carrageenan injection [8]. Individual rats were placed in an acrylic box with a raised wire mesh (Anesthesiometer; UGO BASILE, Varese, Italy) and was allowed to acclimate to the new environment for 5 min. Withdrawal reflexes responding to mechanical stimulation on the surface of the rat’s scrotum and tail base were tested using von Frey filaments. This could generate a reproducible stimulus strength in grams (marking force of 0.02–0.60 g) with different sizes of filaments (ranging from 0.236 to 0.384 mm in diameter). Stimulation with filaments was applied ten times at 10-s intervals alternately to the ventral surface of the scrotal wall and to the tail base after waiting for 3 min. The response threshold was analyzed in three out of five applications that generated reflex withdrawal.

### 4.7. Immunohistochemistry

Prostate inflammation was estimated based on the results of immunohistochemistry in consecutive ventral prostate sections from rats sacrificed 5 days after carrageenan injection. Paraffin-embedded tissue sections of 7 µm in thickness were treated with xylene, dehydrated in ethanol with decreased concentrations, and blocked for endogenous peroxidase activity by 3% hydrogen peroxidase in methanol for 10 min. The tissue sections were incubated in a 10 mM sodium citrate buffer (pH 6.0) for 5 min at 95 °C. The final step was repeated using a 10 mM sodium citrate solution (pH 6.0). The sections were cooled in the same solution for 20 min and then rinsed in PBS. The sections were then immersed with a primary antibody (1:100 dilution) for 1 h at 37 °C. The primary antibodies included: IL-1b (1:250), TNF-a (1:200), Caspase-1 (1:500), NALP1 (1:250), NALP3 (1:250), BAX (1:250), and Caspase-3 (1:250). The signal was visualized using an Envision System (DAKO, Carpinteria, CA, USA) for 30 min at 37 °C. 3,3’-Diaminobenzidine tetrahydrochloride (DAB) was used as the coloring reagent and hematoxylin was used as the counterstain. The slides were examined with a slide scanner, Pannoramic MIDI (3DHISTECH Ltd., Budapest, Hungary), and analyzed with Quantity One software.

## 5. Conclusions

In summary, our study shows that local carrageenan exposure leads to ventral prostate inflammation and increased pain sensitivity (allodynia and hyperalgesia) in male rats. These histological and behavioral changes can be modified by ICTA (IL-1b- and TNF-a-targeting aptamers), and the preventive effects may be associated with its ability to neutralize carrageenan-induced pro-inflammatory cytokines and apoptosis activation in interstitial mononuclear cells. Our findings suggest that pro-inflammatory cytokine activation plays a critical role in chronic prostatitis and that targeting these cytokines to initiate apoptosis is a potential strategy for treating pain hypersensitivity caused by local inflammation.

## Figures and Tables

**Figure 1 ijms-21-03953-f001:**
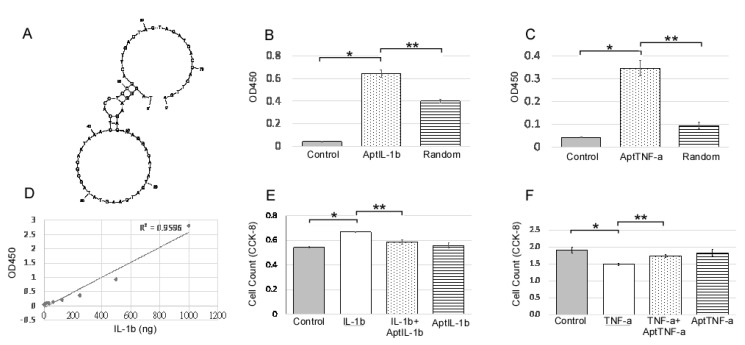
IL-1b-targeting aptamer (AptIL-1b) binds to IL-1b with high affinity and serves as a molecular probe for detecting IL-1b in vitro. (**A**) The predicted secondary structure of AptIL-1b. (**B**) The binding of AptIL-1b and human IL-1b; AptIL-1b binds to human IL-1b fixed in a plate (*n* = 3). (**C**) In vitro detection of the tumor necrosis factor-alpha (TNF-a) signals by TNF-a-targeting aptamer (AptTNF-a) or a random sequence pool in a plate fixed with human TNF-a (*n* = 3). (**D**) Dose-dependent binding of AptIL-1b to IL-1b demonstrated a standard curve in enzyme-linked immunosorbent assay (ELISA) by replacing primary antibody with AptIL-1b. (**E**) The proliferation of the Jurkat cell line detected with Cell Counting Kit-8 (CCK8, Sigma-Aldrich, Product No. 96992) at 48 h after IL-1b with/without aptamer administration (*n* = 3). (**F**) The proliferation of murine urothelial cell line detected with CCK8 at 48 h after the administration of TNF-a with/without aptamer (*n* = 3). The data are presented as mean ± standard error of the mean and were analyzed by Student’s t-test. Asterisks denote statistically significant differences. * *p* < 0.05 represents a significant difference for the CPPS group compared with the sham-operated control group. ** *p* < 0.05 represents a significant difference for the CPPS + ICTA group compared with the CPPS group (*p* < 0.05).

**Figure 2 ijms-21-03953-f002:**
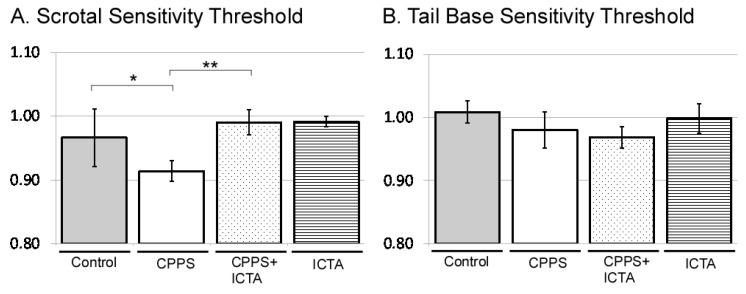
Inflammatory cytokine-targeting aptamers (ICTA) attenuated local carrageenan-induced hypersensitivity of scrotal wall (**A**) and tail base (**B**) allodynia in the von Frey filament test of Sprague Dawley (SD) rats. (**A**) The CPPS + ICTA group compared with the chronic pelvic pain syndrome (CPPS) group (*n* = 6). Local carrageenan treatment resulted in reduction of mean responsive intensity in tail reaction from tactile stimulation in the SD rats. Treatment with ICTA significantly reduced carrageenan-induced pain hypersensitivity in the SD rats. The results are expressed as the mean ± standard error of the mean (SEM) of six animals in each group and were analyzed by one-way analysis of variance (ANOVA). * *p* < 0.05 represents a significant difference for the CPPS group compared with the sham-operated control group. ** *p* < 0.05 represents a significant difference for the CPPS + ICTA group compared with the CPPS group (*n* = 6).

**Figure 3 ijms-21-03953-f003:**
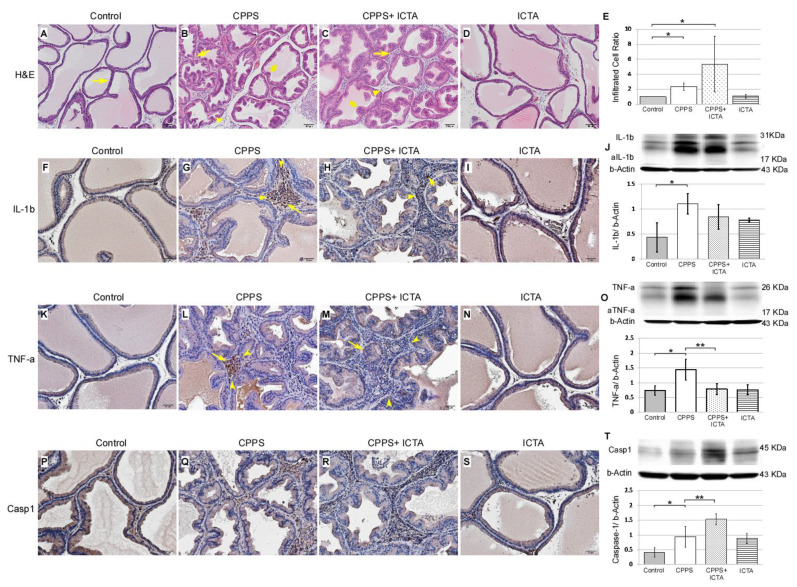
Inflammatory cytokine-targeting aptamers (ICTA) affected carrageenan-induced mononuclear cell infiltration, as assessed by interleukin-1 beta (IL-1b), tumor necrosis factor-alpha TNF-a, and Caspase-1 staining, in the prostate glandular epithelium of the Sprague Dawley rats (SD rats). The scheme used an Haemotoxylin& Eosin (H&E) staining section (**A**–**D**) presenting an overview at 100× magnification. Immunohistochemical staining (**F**–**I**, **K**–**N**, **P**–**S**) at 200× magnification. Representative photomicrographs of IL-1b (**F**–**I**), TNF-a (**K**–**N**), and Caspase-1 (**P**–**S**) immunohistostaining in the rat prostate 168 h after injection. Western blots demonstrated the expression level of IL-1b (**J**), TNF-a (**O**), and Caspase-1 (**T**) over respective beta-Actin. Most of the mononuclear cells were filled in the interstitial parenchyma (**B**,**C**). Carrageenan induced an increase of numerous activated mononuclear cells, some with enlarged cell bodies (arrow) and others with a high nuclear/cytoplasm ratio (arrowhead) (**B**,**G**,**L**,**Q**). ICTA caused a more swollen epithelium in carrageenan-induced glandular hyperplasia (diamond) (**B,C**). Mononuclear cell infiltration was quantified by counting the number of nuclei in the interstitium between the glands (**E**). ICTA treatment resulted in increased mononuclear cell infiltration, but reduced carrageenan-induced IL-1b-related TNF-a-expressing cells in the interstitial space between prostate gland acini (**M**). Although the reduction in IL-1b expression in the CPPS + ICTA group remained insignificant, TNF-a expression was significantly reduced (**M**,**O**). These cells also showed a prominent increase in Caspase-1 expression (**R**,**T**). TNF-a expression was quantified by counting the number of TNF-a positive cells in the prostate glandular epithelium, and by Western blotting analysis of prostate protein extract (**O**). The results are presented as the mean + standard error of the mean (SEM) of six animals in each group and analyzed by Student’s t-test. * *p* < 0.05 represents significance for the CPPS group or CPPS + ICTA group compared with the control group. ** *p* < 0.05 represents significance for the CPPS + ICTA group compared with the CPPS group (*n* = 6).

**Figure 4 ijms-21-03953-f004:**
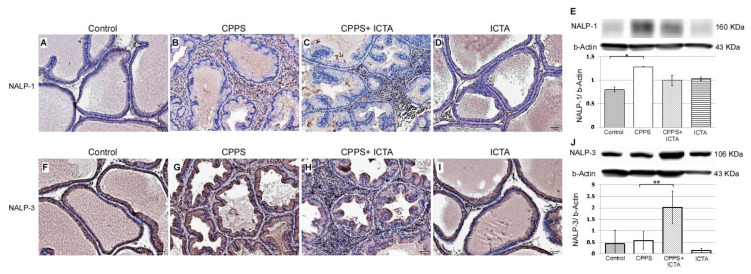
Inflammatory cytokine-targeting aptamers (ICTA) attenuated local carrageenan exposure-induced increases in inflammasome activation (NALP1) in rat prostate, but promoted NALP3 formation 168 h after carrageenan injection. (**A**,**B**) The expression levels of NALP1 and NALP3 following carrageenan injection in the prostate were elevated compared with those in the control group. (**C**,**D**) ICTA attenuated carrageenan-induced increase in the expression levels of NALP1 in the prostate of Sprague Dawley rats, but did not change much when treated alone compared to control. (**E**) Western blots demonstrate the expression ratio of NALP1 over internal control beta-Actin. The ratio is shown in y-axis. (**F**–**G**) Following carrageenan injection, the concentration of NALP1 in the prostate was elevated compared with that in the control group in the Sprague Dawley rats. (**H**) ICTA promoted carrageenan-induced increase in the concentration levels of NALP3 in the prostate of SD rats. (**I**) ICTA alone did not change the expression level of NALP3. (**J**) Western blots demonstrate the expression ratio of NALP3 over internal control beta-Actin. The ratio is shown in y-axis. The results are expressed as the mean + SD of six rats in each group and were analyzed by Student’s t-test. * *p* < 0.05 represents significance for the chronic pelvic pain syndrome (CPPS) group compared with the control group. ** *p* < 0.05 represents significance for the CPPS + ICTA group compared with the CPPS group (*n* = 6).

**Figure 5 ijms-21-03953-f005:**
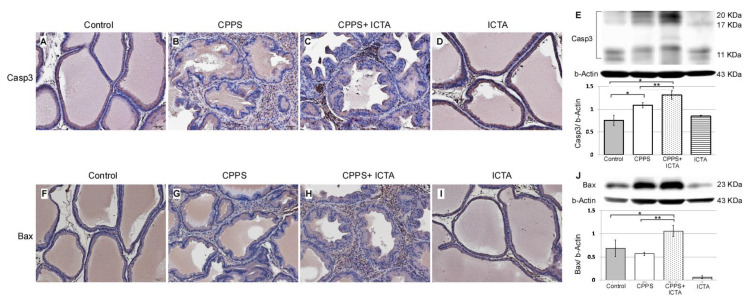
Inflammatory cytokine-targeting aptamers (ICTA) significantly induces apoptosis after neutralizing local interleukin-1 beta (IL-1b) and tumor necrosis factor-alpha (TNF-a) responses in the carrageenan-induced chronic pelvic pain syndrome (CPPS) rat model (*n* = 6). (**A**–**D**) Immunohistochemical staining results show that ICTA prominently promoted pro-caspase-3 expression and cleavage in CPPS prostate (**C**) compared with those in the CPPS only (**B**) and control groups (**A**). Carrageenan injection also resulted in procaspase-3 expression, but this was more pronounced when CPPS was treated with ICTA. (**E**) Western blot results show 106% and 140% increases of activated caspase-3 expression in the CPPS + ICTA group when compared with the CPPS group and control, respectively. (**F**–**I**) Immunohistochemical staining results show that ICTA prominently promoted B-cell lymphoma 2-associated X protein (BAX) expression and cleavage in the prostate of the CPPS + ICTA group (**H**) compared with those in the CPPS group (**G**) and control group (**F**). (**J**) Western blot results show that there were 86% and 54% increases of BAX expression in the CPPS + ICTA group when compared with the CPPS group and control, respectively.

**Figure 6 ijms-21-03953-f006:**
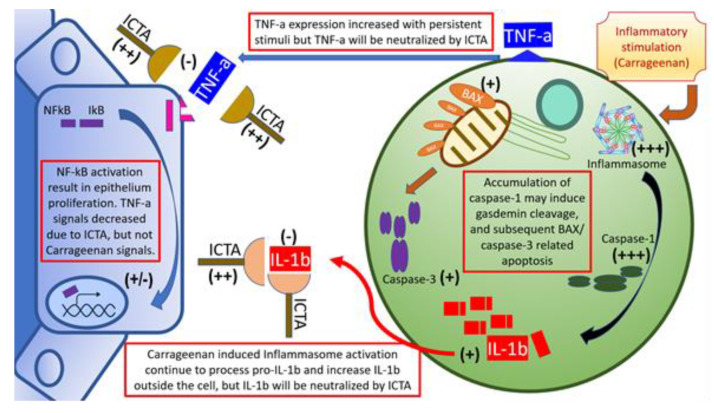
Schematic illustration of inflammatory cytokine-targeting aptamers (ICTA) and interstitial mononuclear cell interaction in the prostate of carrageenan-induced local inflammation. Local carrageenan exposure led to prostate inflammation, including increases of mononuclear cell infiltration activation, and endothelium-related pro-inflammatory cytokines interleukin-1 beta (IL-1b), and tumor necrosis factor-alpha (TNF-a), which induced a positive feedback loop. The above effects may have caused prostate acinar endothelium to undergo hyperplastic change, and interstitial space to become edematous and excessively prominent in the chronic pelvic pain syndrome (CPPS) group, which enhanced pain sensitivity (allodynia and hyperalgesia) in CPPS rats. These compromises in both prostate structure and animal behaviors are affected by ICTA (aptamer neutralizing IL-1b and TNF-a). When cells are overloaded with the active inflammasome pathway, it may induce gasdermin D (GSDMD)-mediated cell death due to caspase-1 overexpression [22]. Cleaved gasdermin can permeabilize the mitochondrial membrane to trigger the mitochondrial apoptotic pathway, which can be demonstrated by the increased caspase-3 cleavage and B-cell lymphoma 2-associated X protein (BAX) expression in the ICTA-treated CPPS group. This suggested that proinflammatory cytokine neutralization may work through inducing apoptosis in inflammatory cells.

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
