# Peer review of "Beneficial Effects of Inflammatory Cytokine-Targeting Aptamers in an Animal Model of Chronic Prostatitis"

_ijms, 2020, doi:10.3390/ijms21113953_

Round 1
Reviewer 1 Report
In this study, the authors investigated the effects of Inflammatory Cytokine-Targeting Aptamers (ICTA) targeted against TNFalpha and IL-1b in a model of carrageenan-induced prostate inflammation in aged rats. Before starting behavioral and cellular experiments, they confirmed proper binding of the aptamers in in vitro experiments. Injection of carrageenan to the prostate induced an inflammatory response with increased levels of mononuclear cells and inflammatory marker associated with pain hypersensitivity. Administration of ICTA inhibited the inflammatory reaction as indicated by reduced levels of proinflammatory cytokines and genes. Furthermore, inflammatory hyperalgesia could be alleviated. In addition, the authors found an increase in apoptosis markers and concluded the neutralization of inflammatory cytokines might result in apoptosis of inflammatory cells.
This is an interesting study concerning effects of aptamers on the regulation of inflammatory responses in prostatitis. However, there are several points that remain unclear and need to be overworked. An important concern is that, although the aptamer effectively bound IL1beta in vitro, there was no significant decrease of IL1b in vivo. This might be a hint that there are problems with delivery and effectivity of the molecule.
Specific points:
- In the introduction, it does not become clear why the authors intended to use the aptamers particularly in prostatitis. It is a relatively general description of inflammatory processes and pain hypersensitivity as well as inflammatory cytokines as potential drug targets in inflammation. This is already well known and several drugs are already approved to directly target these cytokines in inflammatory diseases. So, why is it that important to use aptamers and why in prostatitis?
- The authors only investigated the effects of AptIL1b on cell proliferation. What about AptTNFalpha. Does it also inhibit cell proliferation?
- Until the end of the manuscript it was not completely clear to me what ICTA really is. The authors indicated that both TNFa and IL1b are targeted by ICTA and 2 oligonucleotide sequences are described in the method section. However, it was not clearly described what exactly was injected into the rats.
- The author indicate that they used aged rats but the distinct age was not given. Please add this information to the method section.
- Results, page 4, line 117. It is confusing to describe the effects of CPPS before indicating that this belongs to the CPPS model. I would suggest to first indicate that the CPPs has been used and then describe the results.
- Results Figure 3: The authors indicate a significant decrease of IL1beta after treatment with ICTA in the text (line 123). The diagram does not show this significant difference. Furthermore, concerning the Casp1 staining, it appears that there are a lot of Caspase-positive cells in the control. How can this be explained?
- Figure 4 shows effects 7 days after of carrageenan injection, but the figure legend indicates 144 h.
- It is very difficult to follow the schematic illustration in figure 6. There is no clear flow and the interconnections are not clearly indicated.
- The authors indicate in the method section that they used p-65, p-p65 an IL-18 antibodies but there are no results shown on these genes. However, the involvement of NF-kB is depicted in the scheme and is discussed but not proven by experiments.
- Potential side effects of aptamers and problems with administration should be discussed.
Author Response
Thank you so much for the opinions and suggestions, we tried to do our best to clarify specific points of concern in the following.
Specific points:
Point 1: An important concern is that, although the aptamer effectively bound IL1beta in vitro, there was no significant decrease of IL1b in vivo. This might be a hint that there are problems with delivery and effectivity of the molecule.
Response 1: Our initial study was to apply ICTA through intraperitoneal injection. However, the effect was insignificant. Currently, the application of ICTA through intraprostatic injection is believed to be more straight forward. We believed that because we only block the effect of the end-product in proinflammatory reaction, the production of IL1b is not completely stopped. The amount of IL1b in vivo will depend on the speed of clearance in neutralized IL1b.
Point 2: In the introduction, it does not become clear why the authors intended to use the aptamers particularly in prostatitis. It is a relatively general description of inflammatory processes and pain hypersensitivity as well as inflammatory cytokines as potential drug targets in inflammation. This is already well known and several drugs are already approved to directly target these cytokines in inflammatory diseases. So, why is it that important to use aptamers and why in prostatitis?
Response 2: We choose ssDNA oligonucleotides aptamers for its flexibility and easy production. Aptamers are superior over monoclonal antibodies in their stability for clinical transportation and storage. Therefore, we are testing ICTA in this study.
Prostatitis is a localized inflammatory disease still not well treated in urology. And we had a well-established non-bacterial prostatitis model in our laboratory. Therefore we choose prostatitis.
We had modified the INTRODUCTION to highlight the reason for selecting ICTA and treating prostatitis. “Non-bacterial prostatitis is an inflammatory disease difficult to be treated. Oligonucleotide aptamer is well-known for its stability and flexibility in conjugating varieties of inflammatory molecules.”
And DISCUSSION 3.5 “Although there are already well-known monoclonal antibodies for cytokine neutralization in inflammatory diseases, it may need special transport and storage facilities to apply these drugs in clinical practice. Therefore, we developed ICTA based on single-strand DNA oligonucleotides to meet this need.”
Point 3: The authors only investigated the effects of AptIL1b on cell proliferation. What about AptTNFalpha. Does it also inhibit cell proliferation?
Response 3: AptTNFalpha had been well studied in Reference 20 “Lai, W. Y.; Wang, J. W.; Huang, B. T.; Lin, E. P.; Yang, P. C., A Novel TNF-alpha-Targeting Aptamer for TNF-alpha-Mediated Acute Lung Injury and Acute Liver Failure. Theranostics 2019, 9, (6), 1741-1751.” We also found AptTNFalpha effective in reducing cytotoxicity, instead of inhibiting proliferation, in cell lines. We add Figure 1F and related legends to explain that” (F) The proliferation of murine urothelial cell line detected with CCK8 at 48 h post-TNF-a with/ without aptamer administration (n=3).”. We also add comments on Result 2.1 “As the data further showed that there is a dose-response of AptIL-1b binding to human IL-1b demonstrated with ELISA (Figure 1D), we subsequently investigated in vitro effects of AptIL-1b using the Jurkat cell line (Figure 1E) and AptTNF-a using urothelial cell line (Figure 1F). Our data showed that IL-1b clearly stimulated cell proliferation after 48 h and AptIL-1b counteract its effects. And TNF-a causes cytotoxic effect after 48 h and AptTNF-a counteracts its effect. In contrast, no significant changes were found when treated with AptIL-1b or AptTNF-a alone.” Because we only developed AptIL1b as a new agent in this study, we tried our best to laser focus on AptIL1b to bring new concepts.
Point 4: Until the end of the manuscript it was not completely clear to me what ICTA really is. The authors indicated that both TNFa and IL1b are targeted by ICTA and 2 oligonucleotide sequences are described in the method section. However, it was not clearly described what exactly was injected into the rats.
Response 4: Sorry for the ambiguous description. We had added an explanation of ICTA in Materials and Methods 4.1 Oligonucleotides “ICTA used in the animal study is composed of 150 ug AptIL-1b and 150 ug AptTNF-a in 150 uL PBS for each injection.”
Point 5: The author indicate that they used aged rats but the distinct age was not given. Please add this information to the method section.
Response 5: The actual age of rat in the experiment is 6-week-old. We had noted that in ABSTRACT line 20. However, because clinical non-bacterial prostatitis occurs in all age groups, we replaced the description of ‘aged rats’ with ‘rats’ in the revised manuscript.
Point 6: Results, page 4, line 117. It is confusing to describe the effects of CPPS before indicating that this belongs to the CPPS model. I would suggest to first indicate that the CPPs has been used and then describe the results.
Response 6: Thank you for the pinpoint review. We had modified the sentence to make it more clear “When carrageenan induced CPPS, we found significantly increased numbers of interstitial mononuclear cells, which is an indication of macrophage (arrows) and lymphocyte (arrow head) activation.”
Point 7: Results Figure 3: The authors indicate a significant decrease of IL1beta after treatment with ICTA in the text (line 123). The diagram does not show this significant difference.
Response 7: According to the original data, IL-1b is prominently decreased but not significant. Therefore, we modified the script to “Caspase 1 expression significantly increased in CPPS+ ICTA group (Figure 3T), but the expression of TNF-a in CPPS+ ICTA group was significantly lower compared to that in CPPS group (p< 0.05) (Figure 3O). IL-1b expression was lowered in CPPS+ ICTA group compared to CPPS group, but not significantly (Figure 3J)."
Point 8: Furthermore, concerning the Casp1 staining, it appears that there are a lot of Caspase-positive cells in the control. How can this be explained?
Response 8: The brown color on Figure 3P (Casp1) is actually seen on the mono-layer, cuboidal shape flat epithelium of prostate glands, instead of the interstitial mononuclear cells. The content and character of stain are referred to as background staining by our pathologist. The quantification by Western blotting of prostate tissue extract showed little Casp1 increase in control, which may be more reliable in telling the difference.
Point 9: Figure 4 shows effects 7 days after of carrageenan injection, but the figure legend indicates 144 h.
Response 9: We had corrected the typo. “ICTA attenuated local carrageenan exposure-induced increases in inflammasome activation (NALP1) in rat prostate, but promoted NALP3 formation 168 h after carrageenan injection”. Thank you for pointing out.
Point 10: It is very difficult to follow the schematic illustration in figure 6. There is no clear flow and the interconnections are not clearly indicated.
Response 10: We had modified the illustration by adding explanations for flow and interconnections. Figure 6 legend is modified. “When cells were overloaded with active inflammasome pathway, it may induce gasdermin D (GSDMD)-mediated cell death due to caspase-1 overexpression [1]. Cleaved gasdermin can permeabilize the mitochondrial membrane to trigger the mitochondrial apoptotic pathway, which can be demonstrated by the increased caspase 3 cleavage and BAX expression in ICTA treated CPPS group. This suggested that proinflammatory cytokine neutralization may work through inducing apoptosis in inflammatory cells”. Hope this helps to clarify the proposed mechanism.
Point 11: The authors indicate in the method section that they used p-65, p-p65 and IL-18 antibodies but there are no results shown on these genes. However, the involvement of NF-kB is depicted in the scheme and is discussed but not proven by experiments. Potential side effects of aptamers and problems with administration should be discussed.
Response 11: NF-kB is activated by either Canonical and alternative pathways. We had checked NF-kB expression and found that it might be affected by too many factors in our study design, resulted in large variation among its expression. Besides, inflammation-related cytokines may have cross-regulation between each other [2]. To be more focused, NK-kB and related other cytokine pathways will be studied once aptamers for individual factors are ready. To make an explanation, we added a short comment on DISCUSSION 3.8 Potential problems and research topics with ICTA administration
“There may be more than two inflammatory cytokines activated in response to in vivo carrageenan stimuli. Neutralizing only IL-1b and TNF-a cannot stop everything, but it can demonstrate physiological and pathological changes in this study. However, there are still some problems associated with applying ICTA to inflammatory diseases. First, NF-kB had been evaluated initially in our experiments, but variation among rats is so large that it is hard to pinpoint its role under the current study design. NF-kB had been noted for its complex roles in inflammation that affect both pro- and anti-inflammatory pathways [3], so its role in ICTA administration will need to be clarified. Second, other cytokines not directly associated with inflammasome, such as IL-18, expressed ambiguously in this study (data not shown). Because relationships between pro- and anti-inflammatory cytokine genes can be changed based on the different biological conditions [2], cytokine expression profiles will be an interesting topic for future study when applying ICTA in treating inflammatory diseases.”
- Zheng, Z.; Li, G., Mechanisms and Therapeutic Regulation of Pyroptosis in Inflammatory Diseases and Cancer. International Journal of Molecular Sciences 2020, 21, (4), 1456.
- Kowsar, R.; Keshtegar, B.; Miyamoto, A., Understanding the hidden relations between pro- and anti-inflammatory cytokine genes in bovine oviduct epithelium using a multilayer response surface method. Scientific Reports 2019, 9, (1), 3189.
- Lawrence, T., The nuclear factor NF-kappaB pathway in inflammation. Cold Spring Harb Perspect Biol 2009, 1, (6), a001651-a001651.
Reviewer 2 Report
The study presented by the Authors demonstrates an anti-inflammatory efficacy of IL-1b and TNF-a targeting aptamers that could be used for the treatment of chronic prostatitis. Carrageenan induced prostate inflammation is a well described model of this disease, which is according to a large body of evidence, an autoimmune disorder reducing the quality of life and with very few available therapies. While exact mechanism of the disease is not clear, cytokines such as TNF-α and IL-1β have attracted much attention as mediators in the pathogenesis of chronic prostatitis. The Authors confirmed in the series of experiments that blocking these cytokines indeed alleviates the disease manifestations both at behavioral and histological level.
In section 4.4 the Authors indicate the age of animals writing “arrived in the laboratory on 4th week of gestation” – gestation means pregnancy. Was it the case or the Authors meant 4 week-old rats?
The manuscript needs linguistic and editorial perfecting. There are for example still remaining instructions: the introduction starts with “The introduction should briefly place the study in a broad context and highlight why it is important “. (This is a minor comment).
Author Response
Points in review
Point 1: In section 4.4 the Authors indicate the age of animals writing “arrived in the laboratory on 4th week of gestation” – gestation means pregnancy. Was it the case or the Authors meant 4 week-old rats?
Response 1: Thank you for pinpoint the mistake. The original script” Sprague–Dawley rats arrived in the laboratory on 4th week of gestation” has been modified to “Sprague–Dawley rats arrived in the laboratory on 4th week of age.”
Q2: The manuscript needs linguistic and editorial perfecting.
Response 2: The manuscript had undergone linguistic and editorial perfecting by Enago Editing Services
Q3: There are for example still remaining instructions: the introduction starts with “The introduction should briefly place the study in a broad context and highlight why it is important “. (This is a minor comment).
Response 3: Thank you for pointing out. We had removed the template parts.
Round 2
Reviewer 1 Report
The authors have taken care of all comments of the first review round. Therefore, several points could be clarified. The summary figure has also been improved, however, I still have the impression that it is relatively crowded, complicated and somehow confusing. I would suggest to simplify the figure to make it easier to understand.
Author Response
Point 1: The authors have taken care of all comments of the first review round. Therefore, several points could be clarified. The summary figure has also been improved, however, I still have the impression that it is relatively crowded, complicated and somehow confusing. I would suggest to simplify the figure to make it easier to understand
Reply 1: Thank you for the suggestion.
We had reconstructed the summary figure and hope it can better introduce our finding. Hope the arrangements and annotations can be more precise this way.